# Drivers of the Distribution of Ecological Species Groups in Temperate Deciduous Managed Forests in the Western Carpathian Mountains

**Marian Gabor [1], Pavel Beracko [2], Vladimir Faltan [3], Igor Matecny [3], Lukas Karlik [4], František Petrovič [5], Dusan Vallo [6] and Ivo Machar [7,\*]**

1 National Forest Centre, Institute for Forest Resource and Information, Department of Remote Sensing, Sokolska 2, 96001 Zvolen, Slovakia; marian.gabor90@gmail.com

2 Comenius University in Bratislava, Faculty of Natural Sciences, Department of Ecology, Ilkovicova 6, 84215 Bratislava, Slovakia; pavel.beracko@uniba.sk

3 Comenius University in Bratislava, Faculty of Natural Sciences, Department of Physical Geography and Geoecology, Ilkovicova 6, 84215 Bratislava, Slovakia; vladimir.faltan@uniba.sk (V.F.); igor.matecny@uniba.sk (I.M.)

4 Ministry of Agriculture and Rural Development of the Slovak Republic, Dobrovicova 12, 81266 Bratislava, Slovakia; lukaskarlik1989@gmail.com

5 Constantine the Philosopher University in Nitra, Faculty of Natural Sciences, Department of Ecology and Environmental Sciences, 94901 Nitra, Slovakia; fpetrovic@ukf.sk

6 Constantine the Philosopher University in Nitra, Faculty of Natural Sciences, Department of Mathematics, 94901 Nitra, Slovakia; dvallo@ukf.sk

7 Palacky University, Faculty of Science, Department of Development and Environmental Studies, 17 Listopadu 12, 77146 Olomouc, Czech Republic

\* Correspondence: ivo.machar@upol.cz; Tel.: +420-58-634961

**Abstract:** Managed broadleaf deciduous forests are an important type of forest vegetation in Central Europe, also in the Western Carpathians. These forests are both economically and environmentally valuable. However, little is known about ecological species groups and the inter-specific associations of dominant species in temperate deciduous managed forests in Central Europe. Since the forest stands are in a managed landscape, they are not consistent with the traditionally recognized and used vegetation associations in the Western Carpathians. For these reasons our research contributes to understanding the consequences of broadleaf deciduous forest management. The aim of this research was the determination of ecological species groups and an investigation into the main environmental drivers, in order to explain the distribution of ecological species groups. The numerical TWINSPAN classification was selected to distribute 146 relevés to the five ecological species groups. Of these, 77 relevés were divided into two groups with *Fagus sylvatica* dominant, while 63 relevés were *Quercus petraea* dominant. *Carpinus betulus*, *Tilia cordata* and *Fraxinus excelsior* were dominant in 19 relevés. Constrained Analysis of Principal Components was used to explain the vegetation–environment relationship on three transects in the Male Karpaty Mountains. Altitude, pH, Ca, C, K and Mg were selected as the significant environmental drivers responsible for a large part of the species group variability (31.8%). The main requirement for sustainable forest management is knowledge of the vegetation–environment relationship and this research was focused on gaining such understanding. This knowledge can be used as a decision support tool for sustainable management in managed deciduous forests.

**Keywords:** herbaceous species; classification; soil properties; topography properties; statistical analysis; systematic sampling; Slovakia

## 1. Introduction

Knowledge of species–habitat relationships is important in understanding vegetation patterns in forested landscapes [1]. Those relationships are also an important topic within the framework of the assessment of protected areas and their effectiveness in the conservation of forest vegetation biodiversity [2–4]. In Central Europe, studies on interactions between forest vegetation and environment have been focused on various parameters, such as the niche breadth of tree species under the soil nutrient [5]; forest vegetation and land-use changes [6]; forest vegetation to topographical-soil gradient [7]; beech and spruce forest to soil chemical properties [8]; and environmental controls of plant species richness [9,10].

Deciduous temperate forest vegetation–environment relationships have been studied globally across various patterns along geographic and environmental gradients [11–15].

The existing vegetation is a result of the complex interaction of historic and recent environments and human disturbance factors [16]. Relationships between environment and plant community have often been studied in areas where human disturbance is minimal, such as in protected areas [17]. Only a few studies have taken place in areas with human disturbance [18]. Disturbance may sometimes override the site's influence, especially in forested areas where human disturbances have a long history and play a key role in land-use changes.

Broadleaf deciduous forests are the dominant natural vegetation of Central Europe [19]. The most important broadleaf deciduous forests across Europe, especially Central Europe, are a mixture of either beech and oak forests or beech-dominated forests, and they are important from ecological and socio-economic points of view [20]. This also applies to the forests of Slovakia, which mostly consist of broadleaf deciduous species (63%), although some forested areas at higher altitudes are dominated by coniferous tree species (37%). The dominant broadleaf deciduous species are European beech (*Fagus sylvatica)* and Sessile oak *(Quercus petraea* agg.*)*. Of those forests, 73% are managed and only 23% are not disturbed by human impact (e.g., forests in protected areas).

The concept of ecological species groups is useful in classifying natural communities, determining changes in vegetation, understanding vegetation distribution and environmental factors, estimating species niches, calibrating indicator values for species, modelling the potential distribution of species and plant communities, and assessing habitat quality [21]. The challenge for research into intensively managed forests is to correctly classify forest stands into species groups, because the traditional phytosociological classification was created for forests with a minimal impact from forest management. For this reason, we used the ecological species groups (ESGs) concept as suggested by the study [22]: ESGs are comprised of plants that repeatedly occur together when certain combinations of site factors occur under certain type of forest management. They are species that are perceived to have similar ecological requirements and tolerances as regards environmental stresses and limitations. ESGs are detected by their species composition and abundance patterns among sampling plots. ESG-associated species have similar environmental features. These groups help with the descriptions and the mapping of forest ecosystems in the field through their presence or absence and by the relative coverage of plants in each group.

Forest management consists of anthropogenic disturbances which are able to modulate ecological features [23]. Forest management generally affects abiotic factors in deciduous forests [24], such as microclimate [25,26] and soil nutrients content and dynamics [27]. Tree cutting and coppicing are the most widespread management regimes which affect forest composition, structure and recovery, especially in southern European deciduous forests [28], where mature and old growth forests are rare. On the other hand, growth managed forests in Central Europe occupy a significant area and certain parts of them are both economically and environmentally valuable. Despite the ecological research mainly focusing on the remnants of primeval forests or forest habitats with lower human impact [29–31], some local studies also deal with various aspects of managed temperate forests ecology [32–34]. As indicated in some studies [35,36], there are ESGs of herbaceous species in lowland coppice forests.

Thus, we presumed that other types of forest management also create a basic ecological framework for ESGs of herbaceous species.

The main aim of this study was to investigate the relationship between forest herbaceous vegetation and environment in a forested landscape in the Western Carpathians (Slovakia). In this area, clear-cutting forest management has been in progress since the end of 18th century without any significant changes in land-use to present [37]. This unifying long-term forest management enabled this study to focus on the natural drivers behind and the classification of ecological herbaceous species groups in managed temperate deciduous forests, in order to improve our understanding of the distribution of ESGs as a potential support tool for sustainable forest management.

## 2. Materials and Methods

### 2.1. Study Area

The study area is located in a forested landscape in the Male Karpaty Mountains (Figure 1), which are part of the Western Carpathians in the western part of Slovakia. The vegetation consists of broadleaf deciduous forest with the dominant trees: *Fagus sylvatica*, *Quercus petraea* agg., *Carpinus betulus*, *Tilia cordata* and *Acer pseudoplatanus*. The climate is temperate, the annual precipitation is 760 mm and the monthly range is from 40 mm in January to 80 mm in August, with most precipitation falling in the northwest and the least in the southeastern parts of the Male Karpaty Mountains. The mean annual air temperature is 8 °C and ranges from −2.5 °C in January to 18 °C in July. The prevailing wind direction is northwest [38].

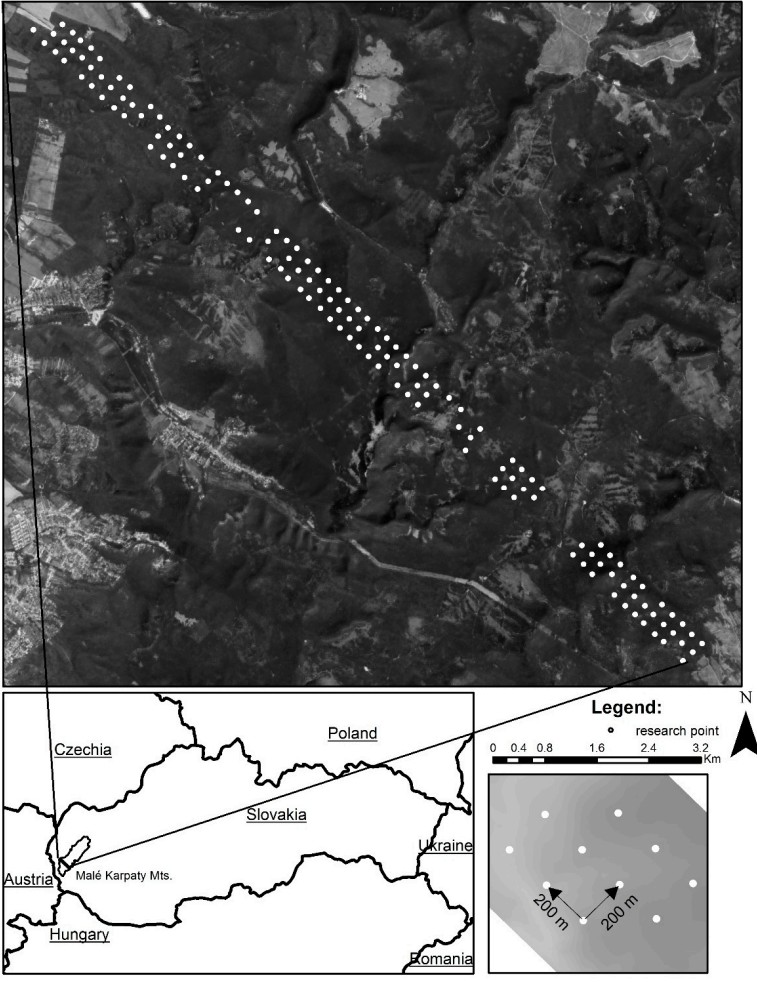

**Figure 1.** Location of the study areas and detailed view of selected study plot.

These forests are influenced by ungulate browsing, which is not sufficiently limited by large carnivores [39]. Forests in the study area have been influenced by local human activities (grazing, litter raking) since the Middle Ages [40]. The first historical forest management plan in study area, based on clear-cutting of mature forests [41], was established in 1761. Currently, all forest stands in the study area are still managed by clear-cutting forest system. Since the forest management practice is unitary in the entire study area, we did not consider it as a variable in our analyses.

## 2.2. Study Plots

We used a random systematic 200 m × 200 m grid sampling plan to establish 186 study plots. Each study plot was 400 m$^2$ in area. They were situated in floristically and ecologically homogenous mature forest stands (Figure 1). The study plots were located on slopes, mid-slopes and summits of hills, at elevations between 200 m and 580 m above see level. The prevailing soil types in the study area are cambisol and rendzic leptosol. The geological substrate predominantly consists of granitic rocks and calcareous limestone.

## 2.3. Vegetation Sampling

The final vegetation database contains, after the removal of 40 study plots (forest cutting during the research period), information about 146 vegetation relevés, with a total of 106 species. In the plots (400 m$^2$) the vascular species in each vegetation layer were recorded. Broadleaf deciduous forests in Central Europe, especially in Slovakia, are characterized by different herb species in summer and in spring. For this reason, we recorded herb species in summer (June and July 2016) and in spring (April and May 2017). The cover of each species was visually evaluated using the Braun–Blanquet approach [42]. Each vegetation relevé was processed in TURBOVEG [43] and JUICE 7 [44] software.

## 2.4. Soil Samplings

In each study plot, soil samples were taken to evaluate the selected chemical characteristics of soil, which can be important as drivers for forest herbaceous species distribution [45]. The soil samples were collected from the upper soil layer, a maximum of 30 cm from the centre of the plot and from two corners (northwest and southeast). Every sample was collected in September, when the weather was stable and chemical properties were also relatively stable. Samples from the corners and the centre were mixed together with a generous one average sample per plot. The samples were then air-dried, homogenized and sieved through a 2 mm mesh. Soil acidity (pH) was measured in KCl (1:2.5 soil/KCL). The total nitrogen (N) was determined by the Kjedahl method. The organic carbon (C) contents were measured by spectrophotometry. The X-ray spectrometry (XRF) technique allowed fast and accurate simultaneous analysis of many elements from each soil sample [46]. The total amounts of aluminum (Al), calcium (Ca), iron (Fe), magnesium (Mn), lead (Pb), potassium (K), sulfur (S), silicon (Si) and zinc (Zn) were measured by the Delta professional XRF spectrometer. Each sample was analyzed five times using the XRF spectrometer, and subsequently the contents of selected variables were calculated by arithmetic means. The main advantages of XRF analysis are the limited preparation required for samples, the non-destructive analysis, the increased total speed and throughput, and the lower start-up and running costs [47].

## 2.5. Topographical Variables

We selected topographical variables, which can explain forest herbaceous species distribution [48]: Altitude (El), slope (Sl), aspect, profile curvature (PrC), plane curvature (PlC), total curvature (TC), potential global year solar radiation (SR), topographical wetness index (TWI), vertical distance to relief channel network (VDChN), slope height (SH), mid slope position (MSP) and topographical position (TPI). In each study plot a topography characteristic was computed in SAGA GIS 2.1.2. The input data for topography characteristics was digital terrain model 1m (DTM) from Lidar data Leica ALS 70/cm. Subsequently, selected topographical variables were derived from DTM. Each topographical variable

was derived in SAGA GIS (2.1.2). Aspect was converted to a southness (South) index and eastness (East) index. The southness and eastness indices were converted by:

$$\phi_{sn} = -\cos(aspect) \tag{1}$$

$$\phi_{ew} = \sin(aspect) \tag{2}$$

Consequently, the values of $\varphi_{sn}$ and $\varphi_{ew}$ range from −1 to 1 and represent the range to which a slope aspects south ($\varphi_{sn} = 1$), north ($\varphi_{sn} = -1$), east ($\varphi_{ew} = 1$), or west ($\varphi_{ew} = -1$) [49].

*2.6. Statistical Analysis*

A modified two-way indicator species analysis (TWINSPAN) [50] was used to classify the 146 relevés into ESGs with similar species abundance patterns. A TWINSPAN analysis was computed in JUICE 7 [44]. Pseudo-species cut levels were set to 0, 5, 25 and 50. Five relevés were selected as a minimum group size for division and a number of ecological species groups were chosen by Sørensen distance and expert view. The fidelity of species to clusters and diagnostic species for vegetation units was calculated using the phi-coefficient based on presence/absence data [51]. In our study, we considered a species characteristic if phi ≥ 0.25. Fisher's exact test ($p < 0.05$) was used to reject the fidelity of species with a non-significant occurrence pattern.

The input vegetation, soil and topography data were measured on different scales. We standardized data before other statistical analysis using a natural logarithm or arcsines.

The non-parametric Spearman rank correlation analysis was applied for testing relationships between all study variables. The significant correlation of some environmental variables, especially between Pb, Zn, Fe, Al, S, TPI and VDChN, were indicated by the correlation matrix (Appendix A). Soil contents of S and Al are closely related to Mg as well as Ca. Therefore, these elements were not used in the model of the residual effect of explanatory variables in the species structure. For the same reason Pb, Fe, TPI and VDCnN were excluded from the following analysis.

Explanation of the structure of the ESG–environment relation was measured using Constrained Analysis of Principal Coordinates (CAP) [52]. The CAP techniques were performed in Vegan package of the R 3.1.0 [53] software environment. The selection of significant factors which created the best ordination gradient was made by forward selection and the Monte Carlo permutation test (999 unrestricted permutations were used). The distance matrix was calculated using the Jaccard coefficient of similarity. Constrained Analysis of Principal Coordinates (CAP) is an ordination method similar to Redundancy Analysis (rda), but it allows non-Euclidean dissimilarity indices. The CAP does not expect multivariate normality, nor does it require a linear relationship between response and explanatory variables. It takes into account the correlation structure in the response data cloud without standardization of the original variables by their variance-covariance matrix. The CAP is a two-step procedure, combining two existing multivariate techniques: classical metric multi-dimensional scaling followed by classical canonical analysis, the Canonical Correlations Analysis on unscaled orthonormal principal componential axes.

The significance of differences among the clusters was tested in combination one-way ANOVA and post-hoc Tukey test in the R 3.1.0 (significant level $p < 0.05$).

## 3. Results

*3.1. Ecological Species Group*

The total of 146 relevés was grouped by TWINSPAN analysis into five ecological species groups (Table 1). The main ESG shows a wide range of variation in stand structure, species composition and affinity to environmental characteristics.

**Table 1.** Synoptic table of percentage frequency (constancy) and fidelity (phi coefficient, upper indices). Only diagnostic species for specific vegetation groups with phi coefficient ≥0.25 are shown in the table.

| No. of ESG | I | II | III | IV | V |
|---|---|---|---|---|---|
| No. of relevés | 29 | 48 | 10 | 19 | 44 |
| *Fagus sylvatica* | $100^{36,6}$ | $98^{34,5}$ | 10 | 79 | 39 |
| *Luzula luzuloides* | $\mathbf{34^{52,0}}$ | – | – | – | 2 |
| *Moehringia trinervia* | $66^{38,7}$ | 2 | – | 5 | $\mathbf{77^{51,5}}$ |
| *Tithymalus amyglaoides* | $31^{27,8}$ | – | 10 | 11 | 11 |
| *Dentaria bulbifera* | $55^{21,0}$ | $\mathbf{63^{29,4}}$ | 10 | 47 | – |
| *Acer pseudoplatanus* | – | $22^{14,6}$ | 20 | 21 | – |
| *Acer campestre* | 3 | 2 | $30^{21,2}$ | $37^{30,8}$ | 2 |
| *Carpinus betulus* | – | – | $90^{54,8}$ | 32 | $64^{27,5}$ |
| *Quercus petraea agg.* | 3 | 4 | $80^{48,8}$ | 32 | $50^{17,1}$ |
| *Tilia Cordata* | – | – | $60^{35,3}$ | $79^{56,3}$ | 2 |
| *Galium sylvaticum agg.* | 3 | – | $\mathbf{50^{63,6}}$ | – | – |
| *Hedera helix* | – | – | $30^{19,2}$ | $47^{43,0}$ | 2 |
| *Impatiens parviflora* | 3 | 6 | $80^{33,5}$ | $84^{37,7}$ | $59^{12,5}$ |
| *Melica uniflora* | – | – | $\mathbf{100^{66,5}}$ | $74^{39,1}$ | 7 |
| *Galium aparine agg.* | – | – | $40^{23,4}$ | $58^{45,4}$ | 7 |
| *Fraxinus excelsior* | – | 2 | – | $42^{56,5}$ | 2 |
| *Corydalis cava* | – | – | 10 | $\mathbf{63^{68,7}}$ | – |
| *Galium odoratum* | 10 | – | 10 | $47^{30,3}$ | $43^{25,3}$ |
| *Geranium robertianum agg.* | 7 | – | 50 | $53^{22,2}$ | $50^{19,4}$ |
| *Ficaria bulbifera* | 10 | – | – | $47^{31,4}$ | $50^{34,6}$ |
| *Alliaria petiolata* | – | – | 50 | $53^{22,3}$ | $57^{26,7}$ |
| *Polygonatum multiflorum* | – | – | 10 | $42^{51,9}$ | – |
| *Anemone ranunculoides* | – | 8 | – | $16^{25,8}$ | – |
| *Calystegia sepium agg.* | 7 | 2 | 30 | 32 | $61^{39,7}$ |
| *Viola reichenbachiana* | 10 | – | – | 11 | $18^{19,3}$ |
| *Mycelis muralis* | 10 | – | – | 26 | $45^{39,2}$ |
| *Cerasus avium* | 7 | – | – | 5 | $16^{22,4}$ |
| *Carex sylvatica* | – | 4 | 10 | – | $11^{14,3}$ |
| *Urtica dioica* | 41 | 2 | 20 | 11 | $75^{49,4}$ |
| *Rubus fruticosus agg.* | 17 | 14 | – | 11 | $50^{40,8}$ |
| *Poa nemoralis agg.* | 3 | – | – | 5 | $16^{25,4}$ |

The first of the five groups was dominated by *Fagus sylvatica*, indicated by *Luzula luzuloides*, *Moehringia trinervia*. *Tithymalus amyglaoides* was included in 29 samples. *Dentaria bulbifera* and *Fagus sylvatica* were the indicator species of the second group, consisting of 48 samples. The third group, with 10 samples, contained *Quercus petraea*, *Carpinus betulus* and *Tilia cordata*. In herb layers *Melica uniflora*, *Galium sylvaticum* and *Impatiens parviflora* were dominant. *Tilia cordata, Fraxinus excelsior, Corydalis cava* and *Polygonatum multiforum* were dominant species in the fourth group of 19 samples. The fifth group was mostly covered by a mix of *Carpinus betulus* and *Quercus petraea* with the dominant herbs *Moehringia trinervia*, *Urtica dioica*, *Mycelis muralis* and *Ficaria bulbifera* present in 43 samples.

### 3.2. Ecological Species Group-Environment Relation

In order to eliminate variables in the CAP we selected six environmental factors (Ca, K, Mg, pH, C and altitude) which significantly affected the gradient of the ordination axes (Figure 2). The variables explained 18.67 (33.92%) of the total variability in the database of the ESG. The first two axes of CAP explained 31.8% of the total variability in the original dissimilarity matrix and 93.7% of all constrained axes. The first of the two axes were correlated with altitude, while the second axis reflected the changes in the chemical parameters of the soil. All CAP models were highly significant (pseudo-F = 11.892 and $p \leq 0.001$).

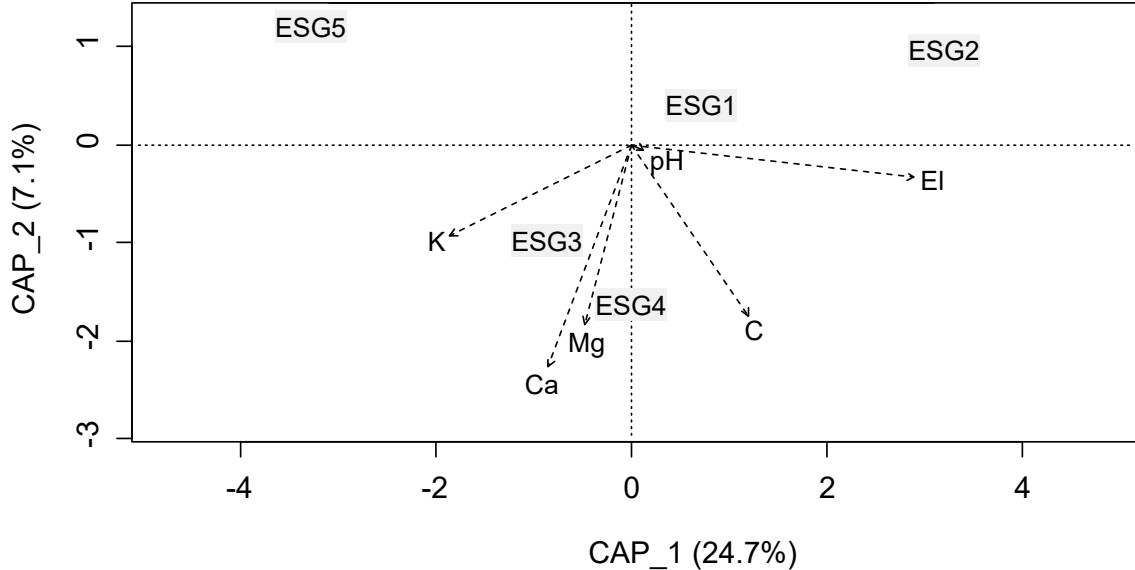

**Figure 2.** Results of a Constrained Analysis of Principal Coordinates showing the position of ecological species groups at the first two restricted ordination axes. The gradients of the ordination axes were modelled by significant environmental parameters selected in a stepwise procedure with backward elimination.

The five ESG were significantly different in their associations with selected environmental variables in relation to soil and topography factors (Table 2). From the details above it is possible to postulate the following: Altitude can be an important factor in distinguishing between ESG 2 (*Fagus sylvatica* and *Dentaria bulbifera*) and ESG 5 (*Quercus petraea agg.* and *Moehringia trinervia*). ESG 5 was localized in the lowest parts of the analyzed area and ESG 2 in the highest. The other ESGs were situated at the same altitude, but differ in soil chemistry. In the first ESG (*Fagus sylvatica* and *Luzula luzuloides*) there was typically the highest soil reaction and also the lowest contents of nutrients (C) compared to the other analyzed groups. ESG 4 (*Tilia cordata* and *Fraxinus excelsior*) can be separated from other ESGs based on the lower mineral content of elements K, Ca and Mg. On the other hand, in ESG 4 it was possible to find the highest content of C. ESG 3 (*Quercus petraea agg.* and *Melica uniflora*) can be described as a group with a lower C content compared to ESG 4, and also because of the higher content of chemical mineral nutrients such as K, Ca and Mg.

**Table 2.** Standard deviation of the most important environmental variables between species groups in the study area.

|  | ESG 1 | ESG 2 | ESG 3 | ESG 4 | ESG 5 | F-Ratio | *p*-Value |
|---|---|---|---|---|---|---|---|
| Ca (%) | 0.412 ± 0.069 | 0.374 ± 0.053 | 0.653 ± 0.117 | 0.572 ± 0.085 | 0.358 ± 0.059 | 2.273 | 0.064 |
| K (%) | 1.494 ± 0.051 | 1.377 ± 0.039 | 1.814 ± 0.085 | 1.692 ± 0.062 | 1.716 ± 0.043 | 12.712 | <0.05 |
| Mg (%) | 0.595 ± 0.031 | 0.577 ± 0.023 | 0.826 ± 0.052 | 0.723 ± 0.038 | 0.615 ± 0.026 | 6.799 | <0.05 |
| C (%) | 1.868 ± 0.199 | 2.435 ± 0.151 | 1.924 ± 0.334 | 2.851 ± 0.242 | 1.509 ± 0.167 | 7.165 | <0.05 |
| pH | 3.723 ± 0.116 | 3.604 ± 0.088 | 3.661 ± 0.195 | 3.498 ± 0.142 | 3.558 ± 0.098 | 0.480 | 0.75 |
| El (m) | 425.752 ± 12.644 | 470.023 ± 9.559 | 287.258 ± 21.159 | 398.017 ± 15.350 | 279.992 ± 10.579 | 52.468 | <0.05 |

## 4. Discussion

In Europe, there have been some classifications of natural forest habitats developed traditionally (since of the beginning of the 20th century). Phytosociological classification (the so-called Zurich–Montpellier approach) is the well-known and widely applying classification system of natural plant communities [42]. Specifically, geobiocenology has been used in Central and East Europe as a classification system focused on natural forests [54]. Recently, CORINE [55] and EUNIS [56] classification systems have been developed. The newest classification system of natural habitats based

on vegetation mapping is a classification system under Natura 2000 European network [57], which is developed in detail in some European countries by creating of national catalogues of natural habitats (in Slovakia, see [68]). Many studies [58–63] have analyzed the impact of various environmental variables on plant species in the natural forests of Slovakia, but only 23% of forests in Slovakia are non-managed natural or semi-natural forests. However, most of our understanding of forest vegetation–site relationships in Slovakia comes from studies performed within forest ecosystems that are relatively undisturbed by humans or from investigations into the effects of particular management strategies on a small scale [64,65]. If we take into account a human impact on forest ecosystems, including land-use change and modifications in the structure of the landscape, it is important to identify the primary driving forces of diversity, not only in undisturbed ecosystems, but also across entire landscapes that have been actively and intensively managed for many years [66,67]. The need for research into the intensively managed forests will grow with the decrease of natural forests in the future, because some of the natural forests in Slovakia are not included in protected areas. In unmanaged forests in strictly protected areas with similar altitude and soil composition, the expected traditional vegetation unit would be predominantly habitats of Acidophilous oak forests, Asperulo-Fagetum beech forests, Luzulo-Fagetum beech forests, and Medio-European limestone beech forests [68]. These vegetation units of forest natural habitats can only be used for mapping and research of natural forests, because the definition of these units is based on both natural structure and species diversity. Our research indicates that when these traditional units of natural forests are disturbed by clear-cutting forest management, it may revert or regenerate into a different vegetation types, such as identified ESGs in the study area.

On the base of TWINSPAN we have identified five species groups and established a list of diagnostic species. This species grouping was not consistent with the traditionally recognized units in the territory of the Western Carpathians [69]. The author of the study [70] found relationships between traditionally used vegetation units in Slovakia and vegetation units created by numerical classifications, but these were in natural or near-natural forest stands. The fact that the numerical classification vegetation units were not consistent with the traditional vegetation units was identified by the study [71]. ESG 1 was statistically similar to *Luzulo-fagion* and ESG 2 was also similar to *Eu-fagenion*, but there were no accurate classifications because some diagnostic species were missing. ESG 3 and ESG 5 were very similar to *Caricipilosae-Carpinenionbetuli*, but ESG 5 had no typical diagnostic species, *Moehringia trinervia*. Only ESG 4 was relatively well classified by floristic composition and ecology as *Tilio-Acerion*.

The main problems with the inclusion of traditional vegetation units were: (1) We did not choose ideal forest stands; (2) we did not carry out research into natural forest stands, but we chose intensively managed forest landscapes, because they are the predominant type of forest in Central Europe, especially in Slovakia; (3) the forest stands have been without the direct influence of management for five years, but they were near the stands where management had a direct influence (logging, path or forest road). An example of this influence was the high amount of *Urtica dioica*, which is not a typical forest herb.

Altitude and soil chemistry played a major role in the ecological species groupings. The highest variation of vegetation was explained by altitude, K and pH [9]. Study of [5] revealed Mg, K and Ca as limiting factors for plant growth in temperature beech forests. These results were expected in accordance with general knowledge on abiotic factors importance in managed forests [72] and local knowledge in Slovakia [73,74].

Altitude had a major role in explaining the variance in vegetation [75,76]. On the high-altitude forest stands only a few species are capable of growth in the characteristically harsh climate, which has lower temperatures, higher wind speeds and more drastic changes in the type and amount of precipitation. Altitude is generally known as an important factor in natural forest vegetation distribution in temperate mountains in the form of forest vegetation zones [77]. But, in managed forests,

vegetation zonation is disturbed by both human activities including forest management practice and climate change impacts [78].

Soil pH played a major role in the separation of ESG 1. It was one of the most important factors affecting the plant community and showed the important role of pH in the separation of species groups [18,21]. The pH only had a minor role in a few studies [79,80]. Soil organic C played a major role in the separation of ESG 4. It had a high absorption capability, which increased the soil´s exchange capacity and therefore its fertility levels. It was an important factor in the variety of deciduous forest in China [81]. Calcium was an important factor in the separation of ESG 3 and ESG 4. Plants use calcium to activate certain enzymes and to send signals that coordinate cellular activities, and it is also responsible for holding together the cell walls. Calcium played a significant role in the separation of species and species groups [58]. Potassium played a major role in the separation of ESG 3. The presence of potassium in the soil makes it easy to transform the water and nutrients in the soil, and it plays a major role in the regulation of photosynthesis, carbohydrate transport, protein synthesis and other phyto-sociological processes. Study [21] showed the importance of potassium in the separation of species groups as well. Authors of the study [82] found potassium as one of the effective factors in the distribution of vegetation types. Magnesium was an important micronutrient in the separation of ESG 4. It is necessary for normal plant growth and has an important role in photosynthesis. Thus, our results are in accordance with the study by [82], which clearly indicated the important role of magnesium in the separation of species groups.

## 5. Conclusions

We have evaluated the relationship between some environmental drivers and vegetation of managed forests dominated by *Fagus sylvatica* and *Quercus petraea* in Slovakia. We found that altitude and soil's chemical properties were effective in the definition of five ecological species groups in the Male Karpaty Mountains. Soil pH, organic calcium, potassium and magnesium were important factors in the separation of individual ecological species groups. Altitude also played a major role in the variance of vegetation in lower mountains of the study area. These results indicate that detailed vegetation research aimed to ecological species groups in deciduous managed forest of Central Europe produces important information for assessment of forest management suitability for herbaceous species and support decision-making in temperate forest restoration, management and planning.

**Author Contributions:** M.G., P.B. and F.P. prepared a study design; methodology was developed by V.F., I.M. (Igor Matecny), and I.M. (Ivo Machar); V.F. and L.K. were responsible for statistical analyses and D.V. was responsible for validation of statistics; field investigation was done by M.G., P.B., V.F., I.M. (Igor Matecny), and F.P.; F.P. and I.M. (Ivo Machar) wrote an original draft; I.M. (Ivo Machar) was responsible for review and editing; project administration was done by M.G. and F.P.

**Funding:** The publication was supported by Slovak Research and Development Agency–grant number APVV-15-0597 Use of geoecological data in the implementation of precision agriculture and, Scientific Grant Agency of the Ministry of Education of Slovak Republic and Slovak Academy of Science–grant number 1/0247/19 Assessment of land-use dynamics and land cover changes and, Cultural and Educational Grant Agency of the Slovak Republic–grant number 032UKF-4/2018 Overview of Methods and Proposal for the Application of the Ecosystem Services Concept in the Environmental Studies Study Program.

**Conflicts of Interest:** The authors declare no conflict of interest.

## Appendix A

**Table A1.** Spearman rank correlation matrix for environmental variables. Red $p < 0.05$; green $p < 0.01$; orange $p < 0.001$.

| | Zn | Fe | Ca | K | Al | Si | S | Mg | C | N | pH | El | Sl | PrC | PlC | TC | TWI | TPI | East | South | SH | SR | MSP | VDChN |
|---|---|---|---|---|---|---|---|---|---|---|---|---|---|---|---|---|---|---|---|---|---|---|---|---|
| Pb | 0.7275 | 0.6911 | −0.0358 | −0.2264 | −0.0807 | −0.3482 | −0.1684 | 0.0367 | 0.6665 | 0.1041 | −0.4277 | 0.7831 | −0.0227 | 0.0269 | −0.0189 | −0.1042 | −0.0848 | 0.1013 | 0.0900 | −0.0253 | 0.1418 | 0.1055 | −0.0297 | 0.2254 |
| Zn | 1.0000 | 0.8751 | 0.2093 | −0.3631 | 0.0169 | −0.1826 | −0.2981 | 0.1204 | 0.4926 | 0.0108 | −0.1334 | 0.5964 | 0.0491 | −0.1022 | −0.0600 | 0.0026 | 0.0503 | 0.0384 | −0.0387 | −0.0700 | 0.2565 | 0.0082 | −0.0377 | 0.1391 |
| Fe | | 1.0000 | 0.1418 | −0.2970 | 0.0808 | −0.2647 | −0.2509 | 0.1389 | 0.5087 | −0.0400 | −0.1554 | 0.6055 | 0.0551 | −0.0553 | −0.0232 | 0.0033 | 0.0737 | −0.0382 | −0.0625 | −0.1080 | 0.1648 | 0.0514 | −0.0043 | 0.1182 |
| Ca | | | 1.0000 | 0.3450 | 0.4706 | −0.4892 | 0.4025 | 0.3078 | 0.2603 | −0.0904 | 0.4103 | −0.3189 | 0.1113 | −0.0383 | 0.0444 | 0.2571 | −0.0239 | 0.1373 | −0.1349 | −0.1941 | 0.0479 | −0.1371 | −0.0149 | 0.1075 |
| K | | | | 1.0000 | 0.6300 | −0.4624 | 0.9661 | 0.4440 | 0.0347 | 0.0128 | 0.1833 | −0.3549 | −0.1377 | 0.0409 | 0.0859 | 0.1680 | −0.0252 | 0.1179 | 0.1195 | −0.0447 | −0.2772 | −0.0310 | 0.0449 | 0.0460 |
| Al | | | | | 1.0000 | −0.1477 | 0.7134 | 0.7081 | 0.1094 | 0.1513 | −0.2373 | −0.3384 | 0.0908 | −0.0057 | 0.0651 | 0.1669 | 0.0231 | 0.1243 | 0.0407 | −0.0359 | 0.0212 | 0.0025 | −0.0717 | 0.0643 |
| Si | | | | | | 1.0000 | −0.4523 | −0.0188 | −0.5817 | −0.1908 | 0.0467 | −0.1812 | 0.2763 | 0.0371 | −0.1072 | −0.1150 | 0.0753 | −0.1649 | −0.0224 | 0.1390 | 0.2080 | 0.0704 | −0.0917 | −0.1608 |
| S | | | | | | | 1.0000 | 0.5011 | 0.0943 | −0.0219 | 0.1656 | −0.3532 | −0.1160 | 0.0467 | 0.1119 | 0.1927 | −0.0394 | 0.1602 | 0.0995 | −0.0341 | −0.2221 | −0.0373 | 0.0203 | 0.0690 |
| Mg | | | | | | | | 1.0000 | 0.0468 | −0.0937 | 0.1158 | −0.1841 | 0.1099 | 0.0012 | 0.1234 | 0.1108 | 0.0163 | −0.0139 | −0.0290 | 0.1776 | 0.0651 | −0.0535 | 0.0501 | 0.0525 |
| C | | | | | | | | | 1.0000 | 0.1012 | −0.4576 | 0.4951 | −0.0988 | 0.0682 | 0.1234 | 0.0560 | −0.0290 | 0.1776 | 0.0651 | −0.0535 | 0.0501 | 0.0525 | −0.0167 | 0.2388 |
| N | | | | | | | | | | 1.0000 | −0.1498 | 0.1807 | −0.3186 | −0.1921 | −0.0626 | −0.1096 | −0.1757 | −0.0802 | 0.0473 | −0.0439 | −0.2707 | −0.1202 | 0.2557 | 0.1083 |
| pH | | | | | | | | | | | 1.0000 | −0.5175 | 0.1485 | −0.0507 | −0.0694 | 0.2624 | −0.0768 | −0.0129 | −0.1274 | −0.1848 | 0.0072 | −0.2584 | 0.0814 | −0.0787 |
| El | | | | | | | | | | | | 1.0000 | −0.1397 | 0.0272 | 0.0579 | 0.0825 | 0.0557 | 0.1585 | 0.2818 | −0.0150 | 0.1520 | | | |
| Sl | | | | | | | | | | | | | 1.0000 | −0.0079 | −0.1276 | 0.3536 | −0.1355 | −0.2181 | −0.1003 | −0.1127 | 0.1903 | −0.1513 | −0.1281 | −0.0240 |
| PrC | | | | | | | | | | | | | | 1.0000 | 0.2376 | 0.2901 | −0.0903 | −0.0057 | 0.0811 | 0.0318 | 0.1891 | 0.0734 | 0.0301 | 0.1084 |
| PlC | | | | | | | | | | | | | | | 1.0000 | −0.2232 | −0.1191 | 0.1554 | −0.0844 | 0.0488 | 0.0900 | 0.0174 | −0.0200 | 0.0223 |
| TC | | | | | | | | | | | | | | | | 1.0000 | −0.1813 | −0.1406 | −0.0162 | −0.0140 | −0.1918 | −0.2956 | 0.0095 | 0.0159 |
| TWI | | | | | | | | | | | | | | | | | 1.0000 | −0.0938 | 0.0166 | −0.1005 | 0.0266 | 0.3862 | −0.1172 | −0.1858 |
| TPI | | | | | | | | | | | | | | | | | | 1.0000 | 0.0709 | 0.2313 | 0.5728 | 0.0802 | 0.0763 | 0.6302 |
| East | | | | | | | | | | | | | | | | | | | 1.0000 | −0.1787 | 0.2024 | −0.1752 | −0.0976 | |
| Sout | | | | | | | | | | | | | | | | | | | | 1.0000 | 0.0727 | 0.0785 | 0.0268 | −0.0058 |
| SH | | | | | | | | | | | | | | | | | | | | | 1.0000 | 0.1088 | −0.0846 | 0.4594 |
| SR | | | | | | | | | | | | | | | | | | | | | | 1.0000 | −0.2085 | −0.0650 |
| MSP | | | | | | | | | | | | | | | | | | | | | | | 1.0000 | 0.1340 |

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
