# Peer review of "Drivers of the Distribution of Ecological Species Groups in Temperate Deciduous Managed Forests in the Western Carpathian Mountains"

_forests, doi:10.3390/f10090798_

Round 1

Reviewer 1 Report

This manuscript presents the results of a study attempting to develop a forest community classification system for managed forests in Central Europe. The authors examined data from 146 sampling units using indicator species analysis to describe 5 distinct ecological species groups. The authors then used Principal coordinates analysis to examine the primary abiotic gradients underlying those communities. This paper takes an interesting approach at a worthy problem (applying systematic ecological study to managed forests when most studies of this type are applied to “undisturbed” forests which make up a relatively small proportion of forests), but is in need of revision prior to acceptance for publication. Specific comments are made below.

Abstract

No significant issues

Introduction

P2, L49-55: Summarize the literature relevant to the work being done in this particular study. An exhaustive list of all forest ecology work conducted in Central Europe isn’t particularly germane.

P2, L56:58: Consider simplifying and incorporating into the previous lines summarizing the existing body of vegetation-environment linkages (e.g., “Deciduous temperate forest vegetation-environment relationships have been studied globally (citations) across various parameters (citations)”)

P2, L59-60: The opening sentence o the paragraph (“The actual spatial…” is redundant with the second sentence, consider omitting

P2: L 64-65: Clarify the statement “in areas where the differences between these factors are relatively small.” Wat are the differences, and what constitutes “small.”

P3: L89-91: Later in the paper, much space is devoted to the influence of abiotic factors on plant communities through the lens of human disturbance. This would be a good place to summarize the literature on how human disturbance influences abiotic conditions (e.g., Ca is found to be  an important abiotic driver of species composition, how do human management activities affect Ca?). This information may be lacking in the literature, but I’d like to see a justification for why the abiotic variables used in the study were chosen.

Materials and Methods

P5, L139-141: See my earlier comment on the justification for selecting these variables for analysis. Again, explain somewhere why are these particular soil nutrients of interest in this study? Do they drive species composition, are they common confounding variables you wish to account for in later modelling exercises, etc.

P5, L148-151: At some point, cite literature summarizing the importance of these topographic variables in explaining species distributions.

Results

P6, L195-196: “The identified ESG does not correspond well to traditional vegetation units, which are not suitable for intensively managed forests.”

P6, L214-P7,L225: This entire section mixes results and explanation in a way that is not appropriate. Strip the potential explanations and postulations out of this section and merge them where appropriate in the discussion.

P7, L220: Soil “reaction” is not a term I’m familiar with in this context

Discussion

What is known about the previous management or human interventions at the site? Is human impact uniform across the site? If not, site history may be an important driving factor behind the ESG’s found here. A discussion of site history would be appropriate either here in the discussion, or in the Materials and methods section. It seems to be a crucial missing part of this study.

P9, L267-273: Further describe the link between altitude and vegetation composition in unmanaged stands and how altitude’s effects in managed stands are different than natural stands.  Specifically: how are the effects of altitude important in these managed stands versus how they may or may not be important drivers of composition in unmanaged forests?

P9, L274-290: Include further explanations of how human impacts affect the levels of these nutrients in these managed forests. We already know how these nutrients affect plant communities, what’s important here is explaining why this is an important factor in managed forests.

 Author Response

Dear Reviewer, We are very grateful for your valuable comments and suggestions to our manuscript. We believe that we have accepted all of your recommendations (see, please, to the List of our responses and corrected version of the manuscript). Thank you for your effort and time. Best regards, Ivo Machar (corresponding author)

Reviewer 2 Report

I think the paper is interesting in a mapping perspective.  The authors are trying to classify disturbance-type vegetation habitat/vegetation cover types.  This could be useful if theses vegetation covertypes can be mapped over large areas and compared to what the original (undisturbed) habitat type was/could be.  It would provide an index of disturbance in the country.  

Specific comments.

When using the term biodiversity in this paper, make it clear that you are referring only to vegetation biodiversity.  

Beech not beach.

Environment-vegetation relationships have also been described for North America--include some references for that contintent.

When authors indicate that the forests are managed--what do they mean?  Are they managed for timber (e.g., clear-cutting?).  If so, are these 2nd, 3rd growth forests.  They mention coppicing.  Who does the managment?  Are the forests managed by the country government?  Locally?  How old is the forest? Or do the authors mean the forests are disturbed by local human use?

Line 83  "ESG are distiguished  from......what?  

Line 91---what are 'high' forests---perhaps mature is a better word? 

Line 110---is hunting of ungulates permitted?  Is overgrazing and lack of regeneration a problem?

Not sure about the term reveles---perhaps this is appropriate for Europe.

Line 181----I think they did a Principle Components Analysis --which is more typically abbreviated PCA.  

Line 195---Can the authors provide some detail about the 'traditional' vegetaion units?  What are those?  Who developed them?  I think this is important because there approach is worthwhile, in part, because it permits a comparison of the authors' findings to these more traditional forest/vegetation units. 

A little more about the land use history at these sites could be helpful if that information is available.  How does the use history (e.g., logging, etc.) potentially affect the physical attributes of the site.  

Does aspect and percent slope affect vegetatiion unit composition?

Line 255-- what do the authors mean by 'close'---statistically similar? 

Line 249---see comment for Line 195 above.  This is an important application of this paper.  The authors could include a paragraph that is something like this:  

"For example, in undisturbed/unmanged protected arveas with similar altitude and soil composition, the expected traditional vegetation unit would be....Our research indicates that when this traditional unit is disturbed, it may revert or regenerate into a different vegation type dominated by species x, y, z."

I think a paragraph that really clarifies how this work can be applied is very important.  

***I did not check formatting nor did I check the references although all 58 mentioned in text seem to be present****

Author Response

Dear Reviewer, We are very grateful for your valuable comments and suggestions to our manuscript. We believe that we have accepted all of your recommendations (see, please, to the List of our responses and corrected version of the manuscript). Thank you for your effort and time. Best regards, Ivo Machar (corresponding author)

Reviewer 3 Report

Detailed comments were included in the text (as comments added in the .pdf file). However, in relation to the entire text, I have one main reservation, the explanation of which I did not find in the content of the manuscript. The authors in this manuscript assumed that the studied forest is a managed one (which is understandable), and that it differs from the natural forest, but they did not refer to the "traditional" classification and did not analyze these differences and their specific reasons. The lack of this reference seems to me a disadvantage of this work. As a result, the purpose of this research becomes unclear, because in fact we do not know why exactly the authors had to do their work. Both in “Introduction” and in “Discussion” it follows, that the authors in their work dealt with the influence of such natural factors as the height and composition of the soil, but by definition they did not include factors directly resulting from forest management. Forest management directly influences the age, species and spatial structure of the stand, and such factors as cutting age, cutting rules, size of areas after felled trees, distance from logging areas, are the factors that shape economic stands. Therefore, I believe that the authors' use of only natural factors affecting stands should be convincingly explained in the chapter "Introduction". Otherwise, the work is not entirely clear.

Author Response

(The authors gave the same response as above.)

Round 2

Reviewer 2 Report

I approve of the changes.